# A Semantic Framework to Support AI System Accountability and Audit ⋆

Iman Naja[1], Milan Markovic[1], Peter Edwards[1], and Caitlin Cottrill[2]

[1] Computing Science, University of Aberdeen, Aberdeen AB24 3UE, UK
[2] Centre for Transport Research, University of Aberdeen, Aberdeen AB24 3UE, UK
{iman.naja, milan.markovic, p.edwards, c.cottrill}@abdn.ac.uk

**Abstract.** To realise accountable AI systems, different types of information from a range of sources need to be recorded throughout the system life cycle. We argue that knowledge graphs can support capture and audit of such information; however, the creation of such accountability records must be planned and embedded within different life cycle stages, e.g. during the design of a system, during implementation, etc. We propose a provenance based approach to support not only the capture of accountability information, but also abstract descriptions of accountability plans that guide the data collection process, all as part of a single knowledge graph. In this paper we introduce the SAO ontology, a lightweight generic ontology for describing accountability plans and corresponding provenance traces of computational systems; the RAInS ontology, which extends SAO to model accountability information relevant to the design stage of AI systems; and a proof-of-concept implementation utilising the proposed ontologies to provide a visual interface for designing accountability plans, and managing accountability records.

**Keywords:** AI · Provenance · Accountability · Ontology.

## 1 Introduction

Artificial Intelligence (AI) solutions are increasingly being deployed in diverse domains such as finance, law and healthcare. However, this widespread adoption does not come without risks and AI systems are increasingly linked to grievously erroneous, unintended or even undesirable behaviours (e.g. perpetuating racism and sexism) [25]. Naturally, then, there is a desire to introduce accountability measures for such systems; and over the past decade this has attracted considerable attention from developers and researchers [6,14], professional bodies [1,25], as well as regulators and policy makers [12,24].

For the purpose of this paper, the term AI system refers to software comprising 'core AI' components (e.g. a machine learning model) and other supporting

---

⋆ Supported by the award made by the UKRI Digital Economy programme to the RAInS project (ref: EP/R033846). The authors acknowledge Jatinder Singh and Richard Cloete for their involvement in the early stages of the Accountability Fabric's design.

functions (e.g. API wrappers) [19] allowing it to function either as a standalone solution, or as a part of a larger system. We consider the development and use of an AI system in terms of four high-level life cycle stages: *Design*, *Implementation*, *Deployment*, and *Operation*; this conforms to the recommendation by Amershi *et al.* [2] that standard software engineering practices should apply to such systems. *Design* involves all aspects associated with designing an AI system; *implementation* encompasses all activities associated with building and testing the system; *deployment* includes installing and configuring the system and, if applicable, integrating it with other systems, producing documentation, and training users. Finally, *operation* consists of the actual use of the system and (routine) monitoring. Moreover, by accountability, we mean the ability to inspect, review or otherwise interrogate an AI system with the goal of (*i*) making processes associated with each of its life cycle stages transparent [1, 6, 7, 14, 24, 25]; (*ii*) demonstrating compliance with hard laws (i.e. laws and regulations), and soft laws (i.e. standards and guidelines) [14, 24]; and (*iii*) aiding investigations into the cause(s) of failure or erroneous decisions and supporting the identification of responsible parties [1, 6, 7, 14, 24].

To realize accountable AI systems, different types of information from a range of sources need to be recorded throughout the system life cycle. We argue that knowledge graphs can support accountability of AI systems by capturing and linking critical transparency information across the different life cycle stages. However, such transparency information must be meaningful and its collection must be proactive (i.e. planned) so it can be enforced through the means of hard and soft laws. We introduce the concept of *accountability plans*, which represent the information that should be captured at different stages of an AI system's life cycle. Accountability plans are linked to *accountability traces*, which are records representing the actual manifestation of those plans. These traces capture structured information describing crucial outcomes of activities influencing the accountability of the AI system. Such activities may represent, for example, the creation of tangible artefacts (e.g. design specifications, implemented system components) or decisions made by key staff members (e.g. approving a design specification) during the system life cycle. Similar to the idea of model cards presented by Mitchell *et al.* [20] which is gaining popularity in the machine learning (ML) community, the instances of "accountable outputs" produced by activities recorded in the accountability traces may be understood as reports or cards detailing the key accountability information. To model *accountability plans* and *accountability traces*, we rely on a provenance-based approach by reusing the W3C recommendation PROV-O [16] and its extension EP-Plan [17]. We extend PROV-O's concepts *entity*, *activity*, and *agent* to represent the *accountability traces* as causal provenance graphs and EP-Plan's concepts *step* and *variable* to describe abstract plans corresponding to such provenance records.

In this paper, we focus on exploring the feasibility of our proposed approach through exploring the core mechanisms for capturing accountability plans and their corresponding traces. We then evaluate this idea by implementing a proof

of concept software tool for documenting the design stage of AI systems which incorporate ML systems. Specifically, our three main contributions are:

1. The System Accountability Ontology (SAO), a generic, reusable, lightweight core ontology which introduces a set of concepts to model accountability plans and their corresponding traces to support accountability of computational systems.
2. The Realising Accountable Intelligent Systems (RAInS) ontology, an extension of SAO, for supporting accountability during the design stage of AI systems, specifically those which employ machine learning.
3. The *Accountability Fabric*, a proof-of-concept implementation utilising SAO and RAInS to provide a visual interface for designing accountability plans, and managing accountability records.

The remainder of this paper is organised as follows: Section 2 discusses related work; Section 3 describes the methodology used when creating the SAO and RAInS ontologies; Section 4 discusses the knowledge representation requirements influencing the design of SAO and RAInS; Section 5 describes SAO and RAInS; Section 6 discusses the implementation of the *Accountability Fabric* and an evaluation of SAO and RAInS; and finally, Section 7 concludes the paper with discussion of future work.

## 2    Related Work

The challenge of how to realise accountable AI systems has attracted considerable attention over the past decade. Professional bodies such as ACM and IEEE have published statements and reports listing principles for accountable algorithms and trustworthy AI systems [1, 25]. National and international regulatory bodies have been working to understand and address the implications of AI systems use; and legislation is being developed across a number of jurisdictions, including the UK and the European Union, with a focus on accountability and maintaining ethical principles [8, 12, 24]. Developers and researchers have also been involved in underscoring the need for accountable AI and have proposed a range of methods to address related issues [28]. Many of these involve documenting how AI systems are designed and developed and how they operate [6, 11, 14, 20]. Typically, such approaches include questions or prompts which designers and developers of AI systems need to consider and for which they should document outcomes. This process is largely manual; however, semi-automated approaches such as the Model Card toolkit[3] are also emerging. Ontologies which describe AI systems and processes that lead to their creation have also been proposed (e.g. MEX [9], ML Schema [22], and KBCE [27]). However, tools to support community uptake (e.g. to automatically produce metadata of ML experiments) are still largely missing and possibly hinder widespread adoption[4]. In the same context, PROV-O has been proposed as a means to record the

---

[3] `https://github.com/tensorflow/model-card-toolkit`
[4] `https://github.com/ML-Schema/core/issues/23`

provenance of decisions made by AI systems [4,13]; while this has some similarities with the approach we describe here - it has a much narrower scope. In our work, we utilise PROV-O's concept of a plan, which represents intended steps or actions so that an objective may be realised; however, no detailed vocabulary for representing such plans is provided. Extending PROV-O for documenting plans was originally proposed for the scientific workflow domain, for example, by the ProvOne [3] and P-Plan [10] ontologies. More recently, P-Plan, and its extension EP-Plan [18], has been applied in other domains. For example, Pandit and Lewis [21] proposed an extension of P-Plan for use in the GDPR context, while Markovic *et al.* [17] discussed the role of EP-Plan in increasing transparency of Internet of Things deployments. Both works demonstrate the cross-domain reusability of P-Plan's simple approach to modelling abstract plans as a series of *steps* interlinked through their input and output *variables* into an acyclic graph.

In summary, ontologies have been proposed to describe AI systems and may be used to enhance their transparency. Ontologies extending PROV-O to enable richer descriptions of plans associated with provenance traces have also been proposed. However, to date we are not aware of any approaches that combine these two to address the challenge of accountable AI systems, and therefore deem our approach to be novel.

## 3    Ontology Development Methodology

The NeOn methodology [23] was adopted to guide the process of ontological modelling of *accountability plans* and their corresponding *accountability traces*. Knowledge representation requirements for accountable design of AI systems were gathered from the academic literature, statements and guidelines released by professional bodies, and publications from regulatory bodies. An application use-case from the healthcare domain was also analysed to identify information elements that should be captured as part of *accountability traces*. The use case is based on plans by the Scottish Breast Screening Programme to address a shortage of trained radiologists [26] by examining how a deep-learning image classifier[5] can be used to replace one of the two human radiologists currently required to analyse mammography images. A number of indicative competency questions were collected, which were then grouped via further analysis under six broad themes discussed in Section 4.

A modular approach was chosen to firstly formalise SAO, which contained the core, reusable concepts for modelling *accountability plans* and corresponding *accountability traces* applicable to generic computational systems. This was followed by formalisation of RAInS which extended SAO for the specific domain of AI system design. The ontologies were implemented using Protégé and further evaluated via a proof-of-concept application for generating and managing knowledge graphs described using SAO and RAInS (details in Section 6).

---

[5] `https://www.abdn.ac.uk/news/12398/`

## 4   Knowledge Capture Requirements

Competency questions (CQs) were extracted from existing literature [4–8, 11, 13, 14, 20, 24]. These covered a range of topics relating to AI systems and their development, including documenting requirements for specific components (e.g. ML models) and explanation of automated decision-making. While the literature did not always explicitly link the identified CQs to a specific life cycle stage (e.g. design), we used our experience from the aforementioned medical use case scenario and our own judgement to identify CQs applicable to the design stage of an AI system incorporating ML. The CQs were then transformed into knowledge capture requirements organised under the following themes to identify what should be recorded by *accountability traces*:

1. **System-level information**: the intended purpose of the system [4, 20, 24]; the intended users of the system [5, 7, 20]; and the compliance specifications which apply to the system, i.e. the hard laws that must be followed and soft laws that should be followed [5, 24].
2. **Dataset information**: its characteristics (e.g. size, composition of instances, number of features) [4, 5, 7, 11, 20, 24]; collection method [8, 11]; any associated pre-processing (e.g. sampling, aggregation) [5, 11, 20, 24]; and the tasks for which it should be used [5, 11] and those for which it should not [11].
3. **Model information**: its characteristics (e.g. decision threshold, excluded dataset features) [6, 13, 20]; details related to implementation (e.g. algorithm used) [6, 20, 24]; associated evaluation procedures [6, 20, 24]; and the tasks for which it should [8, 20, 24] and should not be used [20].
4. **Supporting infrastructure information**: the specification of system components which are not 'core AI' but may still be the source of erroneous behaviour of an AI system (e.g. user interface, API wrappers); the characteristics of supporting infrastructure relevant to the accountability of systems such as specification of human agency and oversight mechanisms (e.g. human-in-the-loop or human-in-command [24]); specification of audit mechanisms [8, 24]; and specification of the level of explanations to be provided by the system [5, 8, 24].
5. **Limitations and risks**: the known or expected limitations of the datasets used to train the decision-making models [5, 8, 24] and the resulting models [5, 20, 24]; and the known or expected risks, including biases, associated with datasets [5–8, 11, 14, 24] and models [5–8, 20, 24].
6. **Human decision making and approvals**: who is accountable for the creation of various specifications including the dataset [11, 13], model [20] and supporting infrastructure specifications; who assessed the fitness of the dataset [8, 24], model, and supporting infrastructure specifications against the system's purpose (and how was this done); which hard and soft laws requirements were included in dataset [5, 8, 24], model [5, 24], and supporting infrastructure [5, 24] specifications and who assessed the compliance of such specifications against those laws (and how was this done); and finally who approved the various specifications that influence the later life cycle stages (e.g. implementation stage).

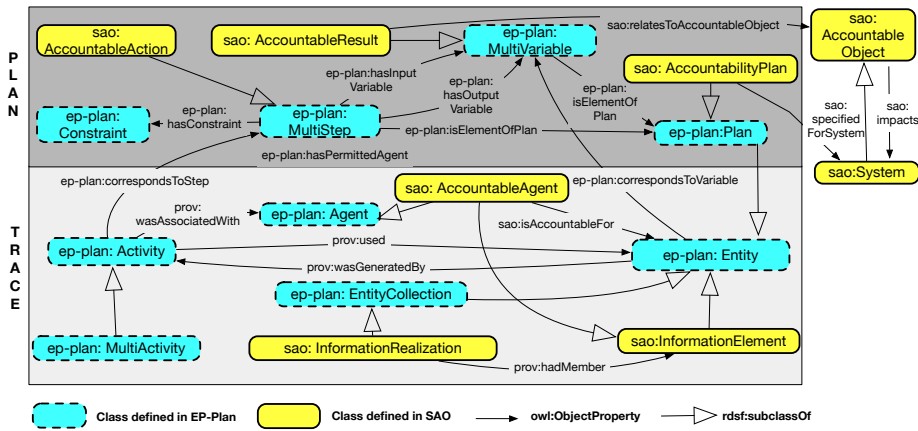

**Fig. 1.** An overview of core concepts defined in the SAO ontology.

## 5   Modelling System Accountability

### 5.1   The SAO Ontology

SAO[6] (Fig. 1) is a generic model for describing accountability plans and corresponding accountability traces to support accountability of computer systems. SAO introduces *sao:AccountableObject* to model an abstract representation of any meaningful grouping (software component, dataset, model, evaluation process, etc.) that may be used to organise system-related accountability information. The definition is deliberately generic so it can be adapted to the needs of any organisation thus allowing flexibility in how a system description should be decomposed into different reference categories that may be used by an audit mechanism. In this context, the system itself (*sao:System*) is an accountable object. A larger system may thus be described as a group of sub-systems or a single system may be broken down into a number of layers/components (e.g. a decision logic layer).

Each instance of *sao:System* may be linked to one or more accountability plans (*sao:AccountabilityPlan*) which specify the information that should be collected to support future accountability. The mechanism for capturing plans and their corresponding execution traces is reused from the EP-Plan ontology [17]. Plans consist of steps which take variables as their inputs or outputs; these are then linked to corresponding accountability traces represented as core PROV-O concepts which are sub-classed in EP-Plan (Fig. 1). SAO extends EP-Plan with two concepts for describing accountability plans: *sao:AccountableAction* and *sao:AccountableResult*, and three concepts to describe the corresponding elements of the accountability trace: *sao:InformationRealization*, *sao:Information-Element* and *sao:AccountableAgent*. An accountable action is any process that

---

[6] https://w3id.org/sao

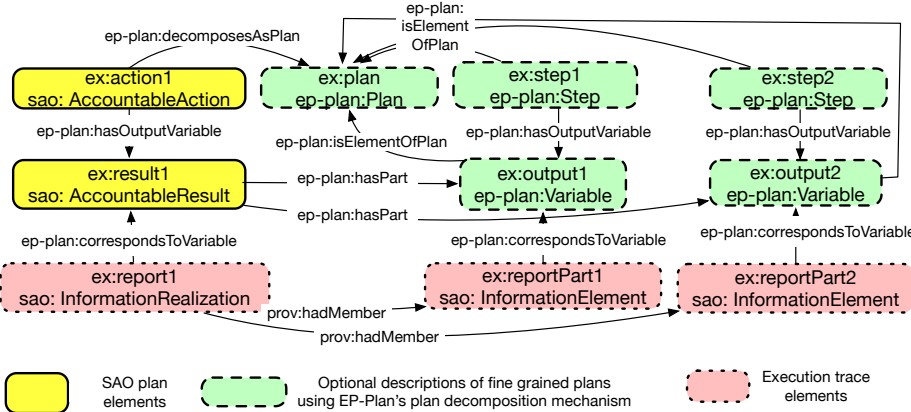

**Fig. 2.** An example decomposition of *sao:AccountableAction* into a sub-plan containing two steps producing variables describing in more detail the composite *sao:AccountableResult* multivariable and matching execution traces.

produces an output (*sao:AccountableResult*) which should be documented for accountability purposes. A description of such an output in the accountability trace then represents a specific snapshot of the information available at a specific point in its production. For example, a specification of an ML model may include characteristics that were believed to be achievable at the design stage (and for which members of the design team were accountable); however, this may differ from the characteristics of the implemented model recorded as an accountable output later in the system life cycle (and for which developers, not designers were accountable). At the accountability trace level, the information corresponding to *sao:AccountableResult* is modelled as a collection (*sao:InformationRealization*). This is because, at the plan level, *sao:AccountableResult* is expected to provide only a high level reference to the expected information. For example, consider a model specification that takes the form of a written report. Here, the plan does not define all the individual steps corresponding to the separate report sections containing the different types of information as output variables (e.g. algorithm details, associated limitations, etc.). Instead, the plan records a high level reference to an *sao:AccountableResult* denoting the expected report. The corresponding execution trace instance is recorded as a collection (*sao:InformationRealization*), which can be linked to any number of *sao: InformationElement*(s) describing the individual records (e.g. algorithm details). The decision to only represent high-level descriptions of plans was made to support the reusability of template plan specifications; thus, steering away from detailed plans which could be more difficult to match to existing internal development processes of different organisations. However, if required, detailed plan descriptions are supported by EP-Plan through descriptions of sub-plans [18] (Fig. 2). This could be utilised, for example, within large organisations where different

agents contribute different information elements and a detailed provenance trace of their individual contributions is required.

SAO also defines *sao:AccountableAgent* to indicate agents that can be held to account for their actions. These are also subtypes of *sao:InformationElement* and may therefore be mentioned as part of an *sao:InformationRealization*. For example, a model specification may specify certain agents that are assumed to be accountable for the realised model. The relationship *sao:isAccountableFor* explicitly denotes the direct link between accountable agents and the entities for which they are accountable. It is also possible to indicate expected responsibilities of agents within an organisation for specific accountable actions and results, by linking *sao:AccountableAgent* to *sao:AccountableAction* using *ep-plan:hasPermittedAgent*.

Finally, the concept *ep-plan:Constraint* can be used to record details about any constraints that were associated with the planned *sao:AccountableAction*, thus providing mechanisms to further customise generic plans to the requirements of individual organisations, allowing further context to be provided on how accountable results were produced (e.g. explaining why certain information elements were included as part of the information realization).

### 5.2   The RAInS Ontology

The RAInS[7] ontology extends SAO for the AI systems' domain by defining a set of concepts required to document the *design* stage of such systems. Figure 3 depicts the classes defined in RAInS.

Subclasses of *sao:AccountableAction* and *sao:AccountableResult* are defined to provide a minimal set of high-level constructs for describing *accountability plans* consisting of actions producing *design specifications* (e.g. a ML model design specification) and *human decisions* (e.g. approval of a specification by an accountable person). By design specifications we mean a collection of requirements or expected characteristics associated with aspects of an AI system. Such specifications are produced by the system designers and should be complied with when the system is realised later in the life cycle. These specifications are not intended to describe specific steps and the order in which they should be performed - i.e. plans. Subclasses of *rains:DesignSpecification* are defined to cover descriptions of dataset, model, evaluation, and supporting infrastructure specifications (see Section 4). Further subclasses of *rains:DesignSpecification* are also defined to describe additional metadata through *rains:SystemPurposeSpecification* and *rains:ComplianceSpecification* to indicate the intended qualities of the expected system, which influence the other individual specifications (e.g. a step producing a dataset specification using inputs defining the system purpose and desired compliance specification). Subclasses of *rains:HumanDecision* are used at the plan level to describe the various decisions (e.g. specification approvals) expected to be made by the accountable decision-makers within an organisation.

---

[7] https://w3id.org/rains.

At the *accountability trace* level, RAInS extends *sao:InformationElement* with a number of subclasses for capturing metadata relating to risks, compliance, intended and incorrect use cases, intended user groups, data collection methods, and data pre-processing methods. Furthermore, information elements may describe a *rains:RealizableObject* which represents a tangible system asset such as a piece of data or software (e.g. training dataset, ML model, component of supporting infrastructure, etc.) that will be realised during the *implementation* stage. Here, the data property *rains:isReusedObject* indicates with a Boolean value whether the resource already exits and is being reused. However, at this stage this system asset is still referred to in abstract terms using *rains:RealizableObject* as it is not yet the implemented AI component. Each *rains:RealizableObject* may be linked using the *rains:hasRealizableObjectCharacteristic* object property to *rains:RealizableObjectCharacteristic*, which may be used to structure the description of *rains:RealizableObject* into separate information elements (e.g. discussing model performance) within *sao:InformationRealization*.

To allow for a wide coverage of potential applications, the RAInS ontology does not dictate what information elements (if any) should be part of an information realization. Users can associate constraints with plan steps that may be used to validate the quality of generated knowledge graphs. Fig. 4 illustrates a SHACL [15] constraint specifying that the *rains:Limitation* element must be present in a collection corresponding to the *rains:ModelSpecification*. Ensuring the completeness of information captured in the knowledge graph would be an important factor, for example, if the collection of *accountability traces* was used to demonstrate compliance with hard or soft laws. If required, constraints may be defined at an abstract level (i.e. they cannot be automatically validated by rules) and their compliance or violation may be determined manually by the information provided by a human agent contributing the relevant *accountability trace* information. This may be implemented via, for example, a question such as "has this activity been performed without any conflict of interests?", where the user is expected to provide a direct answer to this question.

### 5.3   Design Rationale and Alignment to Other Ontologies

EP-Plan (an extension of PROV-O) defines core concepts used for modelling the execution traces corresponding to plan specifications. SAO extends EP-Plan to define concepts for recording accountability plans and corresponding accountability traces for computational systems. RAInS then extends SAO further with domain specific concepts relating to the design stage of AI systems. Accountability plans represent simple and generalisable workflows which document record keeping protocols, whereas much of the *actionable information* is recorded in the accountability traces. This approach is similar to the pattern implemented by the Information Object ontology[8]. Its concept *information object* describes an abstract conceptualisation of an object (e.g. a written text) while the corresponding *information realization* describes a realisation of that object (e.g. a specific

---

[8] `http://www.ontologydesignpatterns.org/ont/dul/IOLite.owl`

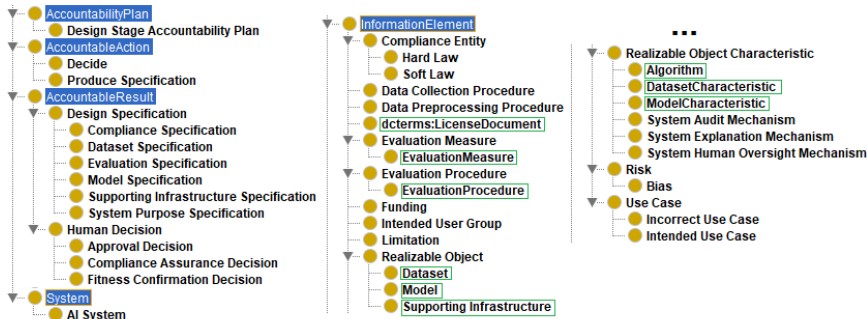

**Fig. 3.** RAInS classes as subclasses of SAO classes (in blue-filled rectangles). Third party classes reused from ML Schema and Dublin Core vocabulary have green borders.

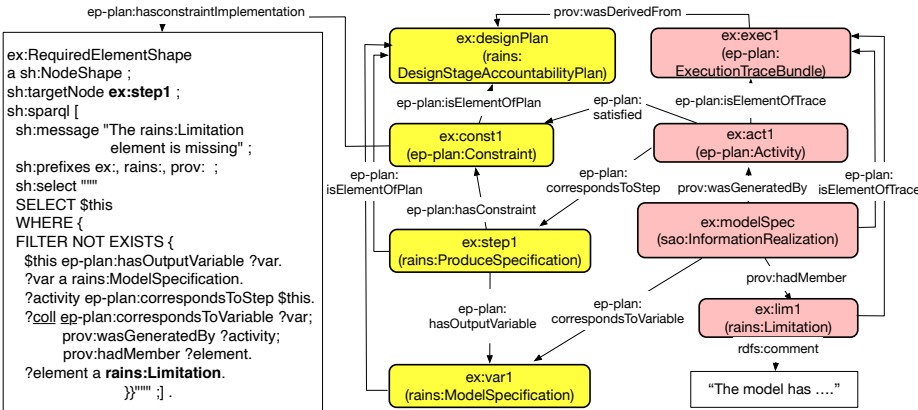

**Fig. 4.** An illustration of linking a SHACL constraint to a plan step using *ep-plan:Constraint*. The constraint states that an instance of type rains:Limitation must be present as part of the *sao:InformationRealization*

report). In our approach, the abstract conceptualisation is the description of *sao:AccountableResult* at a plan level; its subsequent realisation is captured by *sao:InformationRealization*. The latter describes a specific information instance (e.g. a specification report) as part of the accountability trace.

RAInS includes subclasses of *sao:InformationElement* to provide descriptions of information captured in the execution traces. Here, concepts from ML Schema (MLS) and the Dublin Core Vocabulary (DC)[9] such as *mls:Dataset*, *mls:Model*, and *dc:LicenseDocument* are reused as subclasses of *sao:InformationElement*. Classes defined in SAO and RAInS may be extended for more detailed domain specific descriptions. For example, concepts from the Decision Provenance ontol-

---

[9] https://www.dublincore.org/specifications/dublin-core/dcmi-terms/

ogy[10] such as *dp:Question*, *dp:Answer*, and *dp:Option* may be used as subclasses of *rains:InformationElement* to further describe documented human decisions.

## 6 Evaluation

We performed a two stage evaluation[11] process where we first verified the design of SAO and RAInS and then validated their intended application within the accountability context (see Section 1) through prototype implementation and an example knowledge graph.

To support clarity of both ontologies we produced standard documentations using Widoco[12]. The automated OOPS! Pitfall Scanner[13] was used throughout the ontology development process to prevent common pitfalls and bad modelling practices. The scanner did not highlight any issues directly related to SAO or RAINS[14].

We then implemented the *Accountability Fabric* prototype, a web-based tool for managing accountability plans and accountability traces. The tool (available on GitHub[15]) is a *Spring Boot*[16] app with HTML/JavaScript/CSS front end. It comprises three modules: *Accountability Plan Design Module*, *Provenance Capture Module*, and *Audit Module*. The *Accountability Plan Design Module* provides a web interface to design *accountability plans*, using steps and variables defined by SAO and RAInS. The *Provenance Capture Module* is responsible for recording the execution traces and associating them with the accountability plans. This module currently only generates web forms for manual human input, which was sufficient to evaluate SAO and RAInS against our knowledge capture requirements identified in Section 4. However, in future we plan to extend it to enable programmatic access for automated logging (see Section 7). The *Audit Module* provides a simple visual interface which allows the inspection of an AI system's accountability traces. Accountable agents are displayed along with their accountable actions and corresponding accountable results. Lastly, the storage and querying of the knowledge graphs is supported by the GraphDB[17] graph store, SPARQL and RDF4J library[18]. Information is described using PROV-O, EP-Plan, SAO, RAInS, DC and MLS (see Section 5).

---

[10] `https://promsns.org/def/decprov/decprov.html`

[11] Evaluation results and instructions on how to reproduce them are available in the GitHub repository: `https://github.com/RAINS-UOA/ESWC_2021_Evaluation`

[12] `https://w3id.org/widoco`

[13] `http://oops.linkeddata.es/`

[14] The tool produced one incorrect suggestion about potential class equivalence between *sao:System* and *prov:Organisation*. For completeness, we note that the reused ontologies PROV-O, DC, P-PLAN (which EP-Plan extends) and MLS - which we have no control over - produce a number of warnings related to missing domains and ranges, missing inverse properties, etc.

[15] `https://github.com/RAINS-UOA/rains-workflow-builder/tree/ESWC-2021`

[16] `https://spring.io/projects/spring-boot`

[17] `https://www.ontotext.com/products/graphdb/`

[18] `https://rdf4j.org/`

The *Accountability Fabric* was used to create an accountability plan[19] for the design stage of an example ML-based medical image classification system. It was then used to create an accountability trace[20] associated with the plan. Fig. 5 depicts a portion of the generated knowledge graph[21] modelling the accountability plan and its corresponding trace. It illustrates an example where a dataset specification (*ex:2*) is described as an information realization containing an information element (*ex:5*); the latter describes its data collection procedure.

The knowledge graph was imported into Protégé[22] and the built-in Hermit reasoner was used to evaluate the consistency of the populated ontology and to infer additional relationships. The inferences were then inspected manually to validate their correctness. While all the inferences were correct, the Hermit reasoner identified an inconsistency originating from the MLS ontology which was initially imported in RAInS for the evaluation (*owl:Nothing EquivalentTo mls:Experiment*). However, this inconsistency does not affect the concepts reused by RAInS and MLS ontology is not imported by default.

To validate if our ontologies satisfy the three goals of accountability as outlined in Section 1 and the knowledge capture requirements described in Section 4, we used the *Audit Module* user interface to retrieve relevant information associated with the design stage of an AI system. The *Audit Module* presents an agent-centric interface focused on identifying how agents were involved in different aspects of the system design, the decisions they made and the outputs they produced. The *Audit Module* executes SPARQL queries[23] to populate the interface. Fig. 6 illustrates a query used by the tool to retrieve details about the accountable agents who performed accountable actions along with the corresponding results. Fig. 7 depicts a screenshot of the interface driven by the data in Fig. 5; the accountable actions of a selected agent are listed in the Results table. By clicking on the individual values of the results table, more information about the corresponding instance is presented in the *Object Details* window.

Using this interface we were able to demonstrate that a system life-cycle stage can be made transparent by retrieving information about accountable agents, their activities, and accountable results described using SAO and RAINS to provide answers satisfying our knowledge requirements (see Section 4). This directly relates to the first of the three accountability goals discussed in Section 1. However, we also note that both SAO and RAInS are incomplete as they require extensions to cover specific domain applications and other life-cycle stages respectively. To satisfy the second accountability goal we inspected the information

---

[19] The example plan was created with 19 steps, each producing one output variable.

[20] The trace contained: 19 activities (corresponding to the 19 steps); 19 information realizations (corresponding to the 19 variables); 16 accountable agents; and 48 information elements, including the 16 accountable agents.

[21] `https://github.com/RAINS-UOA/ESWC_2021_Evaluation/tree/main/exampleKnowledgeGraph`

[22] `https://protege.stanford.edu`

[23] Relevant Java file `https://github.com/RAINS-UOA/rains-workflow-builder/blob/master/rains-workflow-builder/src/main/java/uoa/web/handlers/SystemRecordManager.java`

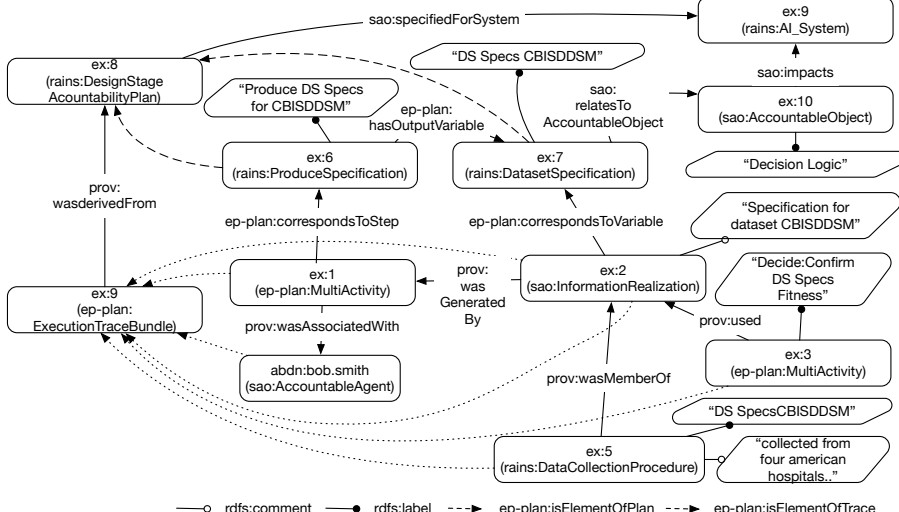

**Fig. 5.** A section of the knowledge graph from the medical image classification example.

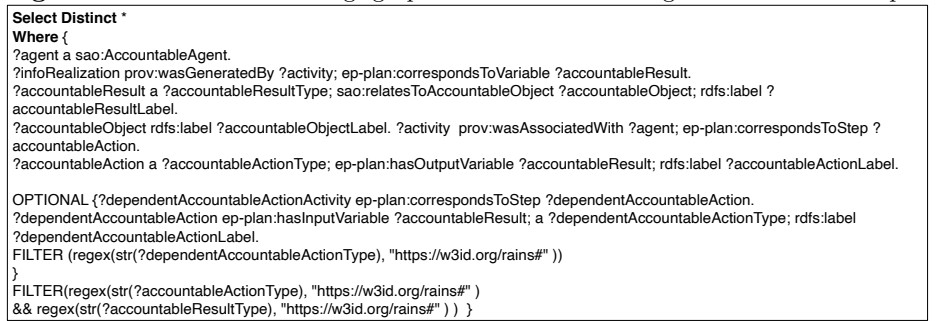

**Fig. 6.** An example SPARQL query for auditing accountability traces.

about human agent decisions (e.g. assuring system's compliance with relevant hard and soft laws). The ability to retrieve such information also demonstrates that the coverage of RAInS in this context is greater than, for example, the reused MLS ontology, and positions SAO and RAInS in a wider context encompassing social, technical and legal perspectives.

To demonstrate the ability of our approach to support identification of errors and responsible parties, consider Fig. 8. Here, it is evident that agents responsible for producing the design specification of the dataset and those responsible for its approval could be potentially held to account for the inappropriate design of the training dataset because of the mismatch between its intended use case and the intended use case of the AI system.

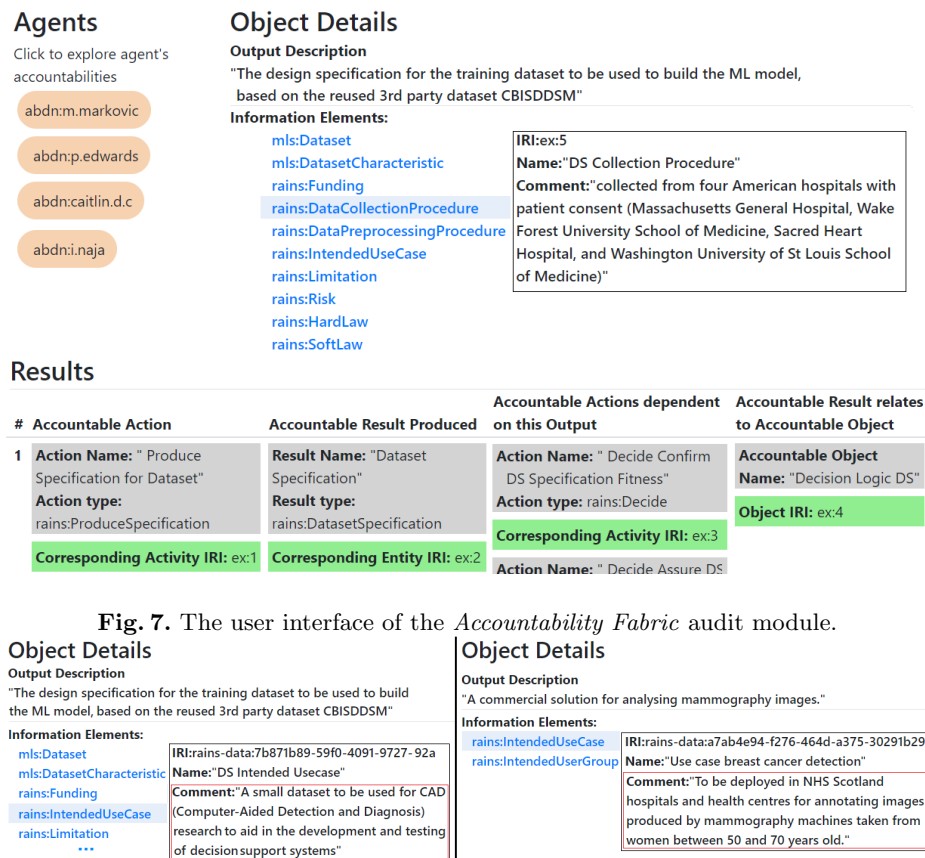

**Fig. 7.** The user interface of the *Accountability Fabric* audit module.

**Fig. 8.** Comparison between the *intended use cases* associated with the reused 3rd party training *dataset* and the overall *system*. The information shows discrepancy between the use cases - i.e. dataset is not suitable for production ready solutions.

# 7    Conclusions and Future Work

In this paper, we have presented an ontology-based approach for supporting accountability of AI systems by increasing the transparency of their design stage using knowledge graphs. We demonstrated, via a proof of concept implementation, the application of SAO and RAInS ontologies to record *accountability traces* by following *accountability plans*.

In our future work, we aim to extend the RAInS ontology with further concepts applicable to other AI system life cycle stages such as implementation, deployment and operation. At the same time, we will expand the functionality of the *Accountability Fabric* framework to evaluate the practical application of the ontology. We also intend to enable data exchange pipelines between the *Accountability Fabric* and external frameworks through API access. For exam-

ple, by integrating with the Model Card Toolkit used by developers to generate model cards [20], the Fabric would be able to extract information related to the implementation of AI systems. Another strand of activity will investigate whether the information contained in accountability plans can be passed to a development environment such as Jupyter Notebook[24] to prevent further model development if accountability information is not provided. Future versions of the *Accountability Fabric* are also set to be evaluated with real users (such as developers of AI systems) to identify real life implications of using such tool. This may include, for example, issues related to commercial sensitivity of AI development if too much information is required by the accountability plan.

Finally, because the *Accountability Fabric* is designed to support collection of information from different sources, we also propose to investigate the challenges relating to the veracity of such information, considering questions such as who created the accountability trace, when was it created, etc. and how emerging standards such as RDF*[25] may help to address them.

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
