# OpenReview forum: "A Semantic Framework to Support AI System Accountability and Audit"
_eswc-conferences.org/ESWC/2021/Conference/Research_Track — ESWC 2021 Research_

### Official Review · AnonReviewer4 · 2021-01-12
**Interesting work however the evaluation is lacking**

**Rating:** 1
**Confidence:** 5
**Impact:** 3
**Design And Technical Quality:** 3

**Review:**

This paper focuses on making artificial intelligence solutions more accountable. Primary contributions include the System Accountability Ontology (SAO), which is a lightweight core ontology used to model accountability plans and their corresponding traces; a Realising Accountable Intelligent Systems (RAInS) ontology, which is an extension of SAO used specifically for the documentation of system design; and a proof of concept Accountability Fabric implementation.

The paper is well written, the research focus is very timely, and the proposed ontologies look very promising. That being said the paper suffers from a number of limitations.

The related work section is very broad focusing more on the high level problem domain than on the potential solutions. I would be interested in knowing more in relation to how the proposal compares to current works in terms of transparency and explainability, especially those that employ semantic technologies.

Additionally, the evaluation of the proposal is very disappointing, as it is not possible to determine the effectiveness of the proposed solution. For instance, I would be interested in knowing more about the use case coverage, such as which uses case scenarios are supported and where extensions would be necessary. I would also be interested in knowing more about the usability of the proposed Accountability Fabric system.

Finally, it is not clear what system engineers or data scientists need to do such that the systems they develop can benefit from the ontologies and the Accountability Fabric system. For instance, how much manual effort is involved and how much could be automated? What is the cost in terms of time and what are the concrete benefits? Is it only possible to record design decisions or can it be used/extended to support auditing of the entire workflow?

*******
*Response to the rebuttal*
*******

Many thanks for the detailed response to my review. Considering the authors response and their offer to provide additional details with respect to the evaluation in the camera ready version of the paper I have elected to raise my overall score to a weak accept.

*******


**Anonymity:**

Yes, I would like my review to remain anonymous.

**Reuse And Availability:**

3: Medium

**Strong Points:**

Strengths of the paper:
- The requirement used to guide the work were compiled from several sources.
- The proposed ontologies and the underpinning ontology development methodology are described in detailed.
- All resources are made available in a public Github repository.


**Subreviewer:**

I submitted this review.

**Weak Points:**

Weaknesses of the paper:
- The related work section is very broad focusing more on the problem domain than on the potential solutions.
- The proof of concept evaluation is very limited, for instance there is no way to assess the generality, coverage, or usability of the proposal.
- It is not clear what system engineers or data scientists need to order to benefit from the ontologies and the proposed Accountability Fabric system.

---

> ### Author Rebuttal · Authors · 2021-01-29
>
> We thank the reviewer for their comments and for highlighting the strengths of this paper in terms of the detailed ontology development description, heterogeneity of sources used to derive requirements, and availability of the research artefacts. We would like to respond to the indicated weak points.
>
>
> 1. The related work: the issue of accountability of AI systems is a lively and, in some respects, controversial topic and hence we felt it was appropriate to provide extensive references to the problem domain. We’ve mentioned existing initiatives such as Model Cards from Google that aim to address a subset of the problem domain we are targeting with our Accountability Fabric. Proposed ontologies such as MEX, ML Schema, and a PROV-based approach proposed by the PLEAD project were also mentioned which could be considered proposals of other solutions in our problem space. We deliberately excluded the existing work on algorithmic accountability and explainability as we deemed it out of scope due to our focus on ascribing accountability to responsible human agents through documentation of AI system lifecycles.
>
>
> 2. The evaluation: as described in the paper, we have performed several “standard” evaluations to verify the design of SAO and RAINs. Further, we regard the implementation of the proof-of-concept framework as confirmation that our models can indeed be used to create practical tools. We therefore feel that the description of the Accountability Fabric was a key element of our evaluation.
> Moreover, while the knowledge graph used in the evaluation may be perceived to be small in size, it is based on our experience from a real-world use case.  Referring to the 3-fold definition of accountability on page 2, we feel that the evaluation demonstrates that our models address all three aspects (but acknowledge that this link could be made clearer in a revised version of the paper):
>
>  i) “making processes associated with each of its life cycle stages transparent”: The competency queries that were implemented to drive the audit manager interface demonstrate the capability of SAO and RAINs to deliver on this point.
>
>  ii) “demonstrating compliance with hard laws (i.e. laws and regulations), and soft laws (i.e. standards and guidelines)”: The knowledge graph, which was produced from the evaluation use case, contains information (accessible via the audit interface), about various regulations that impacted the modelled design processes.
>
>  iii) “aiding investigations into the cause(s) of failure or erroneous decisions and supporting the identification of responsible parties": The knowledge graph  produced in the evaluation does support additional queries that could be added to the evaluation description such as comparison of the system purpose and the intended use of the dataset (i.e., only meant for research purposes) which could demonstrate a potential conflict/bad decision.
> Additionally, as we mentioned in our future work, we aim to perform a user-based evaluation of our proposed approach which could provide further insights on the usability of SAO and RAINS. However, plans for such evaluations have been significantly delayed due to COVID and hence we were unable to report any results before the ESWC deadline.
>
>
> Lastly:
>
> Q: How much manual effort is involved and how much could be automated? In the current implementation of the Accountability Fabric, both accountability plans and accountability traces are created manually. However, the web-based form for creating accountability traces is generated automatically based on the plan description. As we highlight in the future work section, we will aim to expand the fabric with additional data input channels to support semi-automated approaches (e.g., extending the Model Card toolkit).
>
> Q: What is the cost in terms of time and what are the concrete benefits? The cost of time will for sure vary based on the complexity of the system being documented and organisational internal processes. We perceive the main benefits to be: 1) the opportunity of standardisation where we envision that accountability plans and their traces could be potentially shared across organisations, including template accountability plans released by regulators. 2) aligning the representation of the accountability information with W3C standards (RDF and OWL) so that information about individual life cycle stages can be aggregated to support auditing of the entirety of the AI System’s life cycle.
>
> Q: Is it only possible to record design decisions or can it be used/extended to support auditing of the entire workflow? As stated in the future work section, we aim to extend the RAInS ontology with further concepts applicable to the other AI system life cycle stages (implementation, deployment and operation). As the overall vision for the Accountability Fabric is to centralise the accountability information about all these life cycles in its final version it would be indeed possible to audit the “entire workflow”.

---

### Official Review · AnonReviewer3 · 2021-01-12
**Fair contribution**

**Rating:** 1
**Confidence:** 4
**Impact:** 3
**Design And Technical Quality:** 3

**Review:**

The paper describes an ontology to address the problem of accountability of AI systems. The ontology has been described to some detail and made available on-line. A (preliminary) instantiation of it has been done as well. Overall, the paper is a fair contribution and relevant to the conference. However, the validation part needs improvement, as e.g, it is not clear whether the proposed ontology indeed is reasonable for its purpose.

Minor issue:
- sect. 4, point 5.: You implicitly  assume that you have a system whose behaviour depends on machine learning. This may not necessarily be the case, or may be only partially be true, as e.g., you may (also) have KRR-based and/or Multi-agent system where the limitations may involve the knowledge base...

After rebuttal: I acknowledge to have read the authors' rebutal

**Anonymity:**

No, I would like my review to be deanonymized.

**Reuse And Availability:**

4: High

**Strong Points:**

- An ontology addressing the problem of accountability of AI systems.

**Subreviewer:**

I submitted this review.

**Weak Points:**

- the evaluation requires some additional work as it is not clear whether the proposed ontology indeed is reasonable for its purpose.

---

> ### Author Rebuttal · Authors · 2021-01-29
>
> We thank the reviewer for their comments and acknowledgment of our contributions.
>
>
> We agree that the current work described in this paper mainly focuses on ML based systems through the demonstration of SAO’s extension RAINS. Arguably, the ML focus was prioritised based on the current popularity of the field.  However, we would argue that SAO is generic enough to also be applicable in a broader spectrum of systems even outside the ML domain.  Additional clarification clearly stating that the RAINS extension is primarily aimed at ML-based systems will be added to the revised article.
>
>
> As described in the paper, we have performed several “standard” evaluations to verify the design of SAO and RAINs. Furthermore, we regard the implementation of the proof-of-concept framework as confirmation that our models can indeed be used to create practical tools. We therefore feel that the description of the Accountability Fabric was a key element of our evaluation.
> Moreover, while the knowledge graph used in the evaluation may be perceived to be small in size, it is based on our experience from a real-world use case.  Referring to the 3-fold definition of accountability on page 2, we feel that the evaluation demonstrates that our models address all three aspects (but acknowledge that this link could be made clearer in a revised version of the paper):
>
> i) “making processes associated with each of its life cycle stages transparent”: The competency queries that were implemented to drive the audit manager interface demonstrate the capability of SAO and RAINs to deliver on this point.
>
> ii) “demonstrating compliance with hard laws (i.e. laws and regulations), and soft laws (i.e. standards and guidelines)”: The knowledge graph, which was produced from the evaluation use case, contains information about various regulations that impacted the modelled design processes. This information can be retrieved via the audit interface.
>
> iii) “aiding investigations into the cause(s) of failure or erroneous decisions and supporting the identification of responsible parties": The knowledge graph  produced in the evaluation does support additional queries that could be added to the evaluation description such as comparison of the system purpose and the intended use of the dataset (i.e., only meant for research purposes) which could demonstrate a potential conflict/bad decision.
>
>
> Finally, as we mentioned in our future work, we aim to perform a user-based evaluation of our proposed approach which could provide further insights on the usability of SAO and RAINS. However, plans for such evaluation have been significantly delayed by the impact of COVID and hence we were unable to report any results before the ESWC paper deadline.

---

### Official Review · AnonReviewer1 · 2021-01-14
**A good effort to represent accountability in AI systems through agents and workflows**

**Rating:** 1
**Confidence:** 5
**Impact:** 4
**Design And Technical Quality:** 3

**Review:**

## Summary
The paper presents two ontologies: the generic and uppser level SAO describing accountability and its extension RAInS  modelling concepts for AI system - that model provenance information for accountability of AI systems. SAO and RAInS are based on PROV-O and EP-Plan which model ex-ante and ex-post information traces, which permit expressing what is expected to happen and verify if what actually happened is conformant. Along with these, it declares 'steps' and 'agents' associated with accountability which is used to identify points where accountability information is captured/verified and the person/agent accountable. The ontologies are accessible with documentation via GitHub. An evaluation is described using "Accountability Fabric" - a tool to manage accountability plans, where a medical image classification system (ML image classifier).

## Review
The paper outlines ontologies for representing accountable steps/actions and artefacts in AI systems based on provenance workflows, which permits tracking specifications and their implementations along with accountable 'agents'. This is a good effort to systematize the accountability into existing software development processes, and to focus on the 'agent' in addition to recording artefacts for investigations. The authors have taken view of a good selection of relevant literature, and have demonstrated how this can be applied to a ML image classifier use-case. While this work is promising, and certainly will add value and interest to the conference, IMO the work has some questions and issues which need to be addressed (see weak points below). In particular, the evaluation and description of *AccountabilityFabric* should be clarified, and the relation to other sub-fields of AI which may not be well represented as workflows or specifications. Given the venue has a rebuttal stage, I'm open to re-assessing these points based on the author's responses.

## Final comments
The paper takes the view that 'accountability' at the broadest level of abstraction can be reduced to asking, "what did you expect?" and checking if it happened, I think showing this effectively for AI is a challenge both in terms of design and asking the right questions (e.g. auditing). Therefore, I think the work should be presented given its relevance and impact, as well as potential to start discussions.

**Anonymity:**

No, I would like my review to be deanonymized.

**Reuse And Availability:**

4: High

**Strong Points:**


1. Requirements and competency questions are based in existing literature and cover a good breadth of the domain
2. 'AI Accountability' has a working definition, and includes hard+soft laws
3. Abstract upper ontology SAO for representing accountability - good design for extending to other domains
4. Requirements are well represented in terms of different systemic information, risks, and decision making
5. Ontologies are accessible as RDF, documented (detailing concept label+description).
6. The tool, *AccountabilityFabric*, is accessible, and I could build it following the steps listed
7. The KG listed in Evaluation section regarding medical image classification is available as RDF in its entirety. It is based on fictional use-case(s) but relevant towards practical consequences.

**Subreviewer:**

I submitted this review.

**Weak Points:**

1. Citation for AI accountability definition in 1st para of pg.2 (should be addressed in rebuttal or state as original work) - since there are a lot of existing definitions, guidelines, AI ethics documents, and research efforts which deal with accountability.
2. Lack of information or comment about practical use of AI and stakeholders e.g. dataset providers, AI libraries/frameworks, pre-trained or off-the-shelf models. Also lacks information on how this translates to other aspects of AI, such as neural networks / deep learning / feedback training where there are either complex paths (in terms of workflows) or cyclic changes based on output produced (e.g. self correction, backward propogation). I can make probable guesses as to how this might work, but I would prefer this being discussed and acknowledged.
3. (in addition to above) Though the paper targets "AI systems", the concepts and use-case relate specifically to ML systems, which can be a weakness when considering that "AI" is a broad term that encompasses several different types of technologies, some of which would have difficulty in being represented in the given form of workflows. For example, reasoning and logic-based systems may not have as clean a representation of specifications and results as ML classifiers. That being said, this is certainly an open problem within the AI ethics and accountability domain, and its acknowledgement should be sufficient for this work.
4. Footnote 4 is not correctly hyperlinked to https://w3id.org/rains (minor)
5. If the *EntityCollection* is necessary to represent the multiple elements of accountable results that are produced in a trace, then it would be a better design to have the accountable result decompose into *InformationRealization* . Currently, there is *AccountableResult*, which is a subclass of *MultiVariable*, which gets decomposed into *Entity*, which has a subclass *EntityCollection*, which has a subclass *InformationRealization*. Otherwise, there can be an *Entity* which corresponds to *AccountableResult* but is not an *InformationRealization* or *InformationElement*.

So for example, have `sao:correspondsToAccountableVariable` be a sub-property of `ep-plan:correspondsToVariable`. Doing this is not strictly necessary, but if SAO is a generic abstract ontology, it would be better IMHO to have it express these relationships as well to provide a domain-specific vocabulary. (comment, not a point for rebuttal)

5. Why are *AccountableAgent* not represented at the ex-ante or Plan level? For example, expressing that some result must be verified by a specific agent or category of agent? The text does not describe why *AccountableAgent* is a subclass of *InformationElement*? I am confused by the implication that, being a subclass of *Entity*, *AccountableAgent* can be generated by an *Activity*! In the description, the text says, " a model specification may specify certain agents that are assumed to be accountable for the realised model", I presume this refers to use of model specification in the ex-ante or Plan stage given the use of this context in preceding paragraphs. However, the next line describes relationships between *AccountableAgent* and *Entity* via *sao:isAccountableFor*. So how to specify which agent is accountable for a variable? (please address in rebuttal)
6. Is there any concept / method to indicate whether a *Contraint* defined at the Plan level was satisfied/met in the trace? How does this relate to accountability? For example, could one of the tasks of the *AccountableAgent* be to verify the constraints have been met; or to record the results of evaluating/verifying constraints as an *AccountableResult*? (please address in rebuttal)
7. The description and layout of ontology concepts in Table.1 is difficult to read. A diagram would be better for this complex layout, or a 3-column table instead with the last column dedicated to further sub-classes.
8. Footnote 8 describing the acronym RAInS should be in-text for consistency with description of SAO. The acronym should be described at first occurrence. (minor)
9. *AccountabilityFabric*, being listed as a contribution, should have a section of its own, and should not have been described as part of Evaluation.
10. Documentation of ontologies was done using WIDOCO, hence a mention or footnote or citation would be a good step alongside other used tools/references (minor).
11. The GitHub page for the KG describes some details about the use-case and how it is based on existing
12. I found the evaluation section missing crucial details or commentary on whether the use-case and queries satisfied the objectives of the ontology in terms of an accountable AI system? Since the forms used for information input were crafted specific to the ontology, the concepts used to created the KG represent usefulness of the ontology by design. So I'm unsure as to what the evaluation is meant to assess and/or demonstrate. The only conclusion I gathered was that the ontologies could represent the given use-case. However, the follow up query does show one interesting query: who is accountable for what and what were the outcomes. What is missing from this as an 'evaluation' is the description of how this satisfies or helps with the definition of accountability provided in the paper, namely on pg.2. It also leaves the question open as to what other queries and constraints can be utilised based on the concepts provided by the ontologies.
13. The accountability plans should also represent, consider, or explain what happens when something does not meet the required criteria, and needs to be re-developed or re-deployed. While it is possible to consider this accountability at the Plan level by considering the Plan itself as an accountable entity, it can get deeply nested very quickly. E.g. plan checking accountability of previous plan checking accountability of ... I think in real-world, most cases of accountability will be the follow-ups to existing workflows rather than initiation of new systems. Food for thought.

---

> ### Author Rebuttal · Authors · 2021-01-29
>
> We thank the reviewer for their comments. We do not include responses to non-rebuttal items (5,14) and minor ones (4, 8, 9 & 11) here; however, we will address them in the final manuscript. Also point 12 seems to be incomplete.
>
>
> 1) We provide several relevant citations covering the concept of accountability, but now recognise that these were not explicitly linked in the accountability definition on page 2. The definition should now read as: “Moreover, by accountability, we mean the ability to inspect, review or otherwise interrogate an AI system with the goal of (i) making processes associated with each of its life cycle stages transparent [1, 6, 7, 14, 24, 25]; (ii) demonstrating compliance with hard laws (i.e. laws and regulations), and soft laws (i.e. standards and guidelines) [14, 24]; and (iii) aiding investigations into the cause(s) of failure or erroneous decisions and supporting the identification of responsible parties [1, 6, 7, 14, 24].”
>
>
> 2) RAInS provides the concept RealizableObject representing system components which may either be implemented in-house or reused from third parties, e.g. existing datasets, off-the-shelf models, etc. The property isReusedObject indicates that a RealizableObject is a 3rd party resource. SAO’s property isAccountableFor indicates which agent isAccountableFor the RealizableObject. Plans are acyclic but may be linked to different executions, we are currently exploring appropriate mechanisms for cross linking, however, this is out of scope of this work. We expect this to be more relevant for modelling of the implementation stage.
>
>
> 3) We agree that the current work described in this paper mainly focuses on ML based systems through the demonstration of SAO’s extension RAINS. Arguably, the ML focus was prioritised based on the current popularity of the field. However, we argue that SAO is generic enough to also be applicable in a broader spectrum of systems even outside the ML domain. Additional clarification stating that the RAINS extension is primarily aimed at ML-based systems will be added to the revised article.
>
>
> 6) This may be accomplished using EP-Plan, which SAO and RAINS extend. The relationship ep-plan:hasPermittedAgent links an agent to a step and denotes which agents are permitted to perform the corresponding activity. Alternatively, associating ep-plan:Constraint with a step may be used to restrict the identity of an agent or require some identifying qualities (e.g. qualifications) that must be associated with an agent to be responsible for the corresponding activity in the execution trace. In PROV-O, agents and entities are not disjoint; we utilise this when recording agents as part of InformationRealisation which requires its elements to be entities. We chose to omit these details due to space limitations and so as not to confuse the reader with too many details about EP-Plan.
>
>
> 7) This is covered using ep-plan:satisfied and ep-plan:violated relationships (see Fig 3). These relationships may be qualified, and so further provenance may be linked, e.g. describing an activity of constraint evaluation which is associated with a specific agent. We also chose to omit these details for the same reasons as point 6.
>
>
> 10) We regard the implementation of the proof-of-concept framework as confirmation that our models can indeed be used to create practical tools. We therefore felt that the description of the Accountability Fabric was more appropriate as part of the evaluation section.
>
>
> 13) Further to our remarks under 10), we performed several “standard” evaluations to verify the design of SAO and RAINs (and these were discussed in the paper).
> The generated knowledge graph is based on our experience from a real-world use case.  Referring to the 3-fold definition of accountability on page 2, we feel that the evaluation demonstrates that our models address all three aspects (but acknowledge that this link could be made clearer in a revised version of the paper):
>
>  i) “making processes associated with each of its life cycle stages transparent”: The competency queries that were implemented to drive the audit manager interface demonstrate the capability of SAO and RAINs to deliver on this point.
>
>  ii) “demonstrating compliance with hard laws (i.e. laws and regulations), and soft laws (i.e. standards and guidelines)”: The knowledge graph, which was produced from the evaluation use case, contains information about various regulations that impacted the modelled design processes. This information can be retrieved via the audit interface.
>
>  iii) “aiding investigations into the cause(s) of failure or erroneous decisions and supporting the identification of responsible parties": The knowledge graph produced in the evaluation does support additional queries that could be added to the evaluation description such as comparison of system purpose and the intended use of the dataset (i.e. only meant for research purposes) which could demonstrate a potential conflict/bad decision.

---

> > ### Comment · AnonReviewer1 · 2021-02-08
> > **Valuable work, though with shortcomings, should be presented.**
> >
> > I'm satisfied with the responses from the authors. However, I am standing by my original grade of (+1 weak-accept) given that the paper has some lack of clarity about application and evaluation. That being said, I would support its acceptance for presentation as I think the work is a timely contribution to the field, and would add value in future endeavors.
> >
> > (numbers refers to comments in rebuttal response)
> >
> > 1. Accountability definition: I'm satisfied with the solution to base the definition on existing cited sources.
> > 2. I think this point has not been completely answered. The question about reusable components/systems and the complex plans/workflows has been addressed. The comment about applicability to different areas/approaches within AI has not been addressed i.e. whether the solution is feasible or suitable for AI systems other than ML-trainers. IMO this is an important question when concerning "AI Accountability" as a broad topic, and brings about the discussion whether all AI implementations are similar in the sense that a) they can be expressed as a plan; b) have concrete steps that are points of accountability; and c) have measurable (quantifiable) results that can be used to assess in a similar fashion.
> > 3. And then there is also the question of how to represent this as a continuous process (or one that occurs regularly). I believe the use of plans (and provenance) enables all of these at the organisational or managerial level, and that the authors should explicitly state this as a benefit of their approach. I'm satisfied with the solution proposed by the authors to state this as a ML-focused approach.
> > 6. IMO stating association between agents (_AccountableAgent_) and planned steps is an important aspect of provenance to demonstrate, for example, that the organisation or team has an accountability plan in place which features definite responsibilities and roles (in addition to steps and outcomes). This is not about contraints, but about capturing and representing plans of accountability.
> > 7. I'm satisfied with this answer regarding outcome of constraints.
> >
> > (point 11 in review is incomplete, and was attempting to specify relation with existing work / situations)
> >
> > 13. IMO the 'design' of an ontology (evaluated using OOPS, best practice guidelines, etc.) are only one form of evaluation. When an ontology is evaluated via use-case or application, it should also be assessed for suffciency, and suitability. For example, Were there any areas where the ontology was lacking, and this resulted in identification of new concepts/ODP to add or change? This is important, because there are two possible arguments when using an ontology in an use-case: (1) we made do with the ontology we had at hand (demonstrates application), and (2) we used this opportunity to explore how our ontology works or could be made better (shows evaluation). As the authors state in the rebuttal, this could be better explained in the paper in terms of objectives. If it was only about competency question, a simple table showing each CQ was addressed through corresponding concepts would have been sufficient. For applications, it is not sufficient to show 'possibility', since it is difficult to demonstrate. Instead, examples could have been provided - for example regarding queries that investigate something and show that the ontology enables its conclusive answering. This data is present (to a certain degree) in the use-case KG, which also includes various steps, standards (and laws, of different complexity) and IMO the paper can state this more explicitly.
> >
> > Additional comments: it would have also benefited the approach (in general terms) if the paper outlined whether the work would be available in an open fashion (including the accountability fabric?) and how this affords other systems and AI accountability researchers to build/extend this work to their own goals.

---

### Official Review · AnonReviewer2 · 2021-01-14
**A novel framework to support AI system audit based on semantic web technologies**

**Rating:** 1
**Confidence:** 4
**Impact:** 4
**Design And Technical Quality:** 4

**Review:**

Response to the rebuttal

---

I appreciate the author's responses in the rebuttal to clarify the issues that I raised. However, it didn't change my opinion on the paper and I will keep my score.

---

The paper proposes a novel framework to support AI system audit based on semantic web technologies, which aims to address all four stages of AI Systems development: Design, Implementation, Deployment, and Operation, of which the Design stage is currently covered in this paper with RAiNS ontology.

The authors pointed out three contributions of the paper in the form of (i) SAO, a generic ontology to represent System Accountability, (ii) RAiNS, an extension of SAO for the design phase of AI system audit, and (iii) an early prototype of AccountabilityFabric, a prototype for managing system accountability plan and traces.

Pros:

+ The paper is clearly written and easy to understand.
+ It proposes a novel framework to represent and audit accountability plan and traces using semantic web technologies built on their earlier work (E-Plan ontology)
+ The research topic is timely and will encourage future research on its direction.
+ The requirement elicitations and competency questions are based on recent and diverse literature, which are reasonable for the framework.

Cons:

+ The evaluation is currently limited to a qualitative feasibility evaluation with relatively small, manually inputted data. This issue raises questions about whether the proposed framework would work well in a real-world setting (e.g., related to scalability issue, provenance data acquisition).
+ Another weak aspect of the paper is related to the framework's scope, which currently covers only one of four stages of AI Systems. The authors did mention this issue as part of their future work. This issue, however, discourages me from giving a better score for the paper.
+ Furthermore, I tried and failed to run the AccountabilityFabric prototype following the instruction in GitHub, which prevented me from evaluating the prototype further.
+ Minor issues: The ontologies and their respective documentations contain typos and default texts inside.

To conclude, I am excited by this work and looking forward to its further progress, particularly the extension of the framework coverage to include all stages of AI systems and its evaluation with real-world use cases.

**Anonymity:**

Yes, I would like my review to remain anonymous.

**Reuse And Availability:**

3: Medium

**Strong Points:**

* (+) Novelty of the approach
* (+) Strong requirement elicitations.
* (+) Reusable and extensible ontology design.

**Subreviewer:**

I submitted this review.

**Weak Points:**

* (-) Limited evaluation with small, manually curated data
* (-) The current framework only covers one of four stages in the AI lifecycle.
* (-) Not all of the provided artifacts works

---

> ### Author Rebuttal · Authors · 2021-01-29
>
> We thank the reviewer for their comments and acknowledgment of our contributions presented in this paper.
>
> As described in the paper, we have performed several “standard” evaluations to verify the design of SAO and RAINs. Furthermore, we regard the implementation of the proof-of-concept framework as confirmation that our models can indeed be used to create practical tools. We therefore feel that the description of the Accountability Fabric was a key element of our evaluation.
> Moreover, while the knowledge graph used in the evaluation may be perceived to be small in size, it is based on our experience from a real-world use case. Basic statistics about the size and complexity of the generated example knowledge graph (currently available on GITHUB) may be also explicitly described in the paper, which could hint at the scalability of the proposed solution. However, we chose to omit this from the current evaluation description as the size of the graph may vary based on the level of detail recorded and we currently do not have a catalogue of extensive number of real world system descriptions so we could provide at least averaged statistics in this context.
>
> Referring to the 3-fold definition of accountability on page 2, we feel that the evaluation demonstrates that our models address all three aspects (but acknowledge that this link could be made clearer in a revised version of the paper):
>
> i) “making processes associated with each of its life cycle stages transparent”: The competency queries that were implemented to drive the audit manager interface demonstrate the capability of SAO and RAINs to deliver on this point.
>
> ii) “demonstrating compliance with hard laws (i.e. laws and regulations), and soft laws (i.e. standards and guidelines)”: The knowledge graph, which was produced from the evaluation use case, contains information about various regulations that impacted the modelled design processes. This information can be retrieved via the audit interface.
>
> iii) “aiding investigations into the cause(s) of failure or erroneous decisions and supporting the identification of responsible parties": The knowledge graph produced in the evaluation does support additional queries that could be added to the evaluation description such as comparison of the system purpose and the intended use of the dataset (i.e., only meant for research purposes) which could demonstrate a potential conflict/bad decision.
> Additionally, as we mentioned in our future work, we aim to perform a user-based evaluation of our proposed approach which could provide further insights on the usability of SAO and RAINS. However, plans for such evaluation have been significantly delayed by the impact of COVID and hence we were unable to report any results before the ESWC paper deadline.
>
>
> We appreciate the comment regarding the scope of the paper being limited to the design stage. We chose to initially focus on that stage as we found this topic to be underdeveloped in the current literature within the AI accountability context. Furthermore, we were striving in this paper to demonstrate the relationship between our top-level ontology SAO and a vocabulary tailored to a specific life cycle stage, with design being the obvious starting point.
>
>
> We value your efforts to try running the example. We are aware that at least one other reviewer (AnonReviewer1) succeeded at downloading and running the full example as presented on GitHub and in the paper. We would be very happy to amend the instructions in GitHub following further feedback on the encountered issues.
>
>
> Finally, thank you for pointing out that there are typos in the ontologies. We will update the documentation accordingly.

---

### Official Review · AnonReviewer5 · 2021-01-15
**Relevant and challenging research topic - but the accountability concept does not fit**

**Rating:** 1
**Confidence:** 4
**Impact:** 3
**Design And Technical Quality:** 3

**Review:**

Given the authors' rebuttal I have changed the overall score from weak reject to weak accept.

-----------
The topic of this paper is absolutely hot and relevant and any research that investigates the social impact of immersive machine systems should be encouraged and supported.

In the case of this paper, the authors present an interesting approach that might help to better understand AI systems and improve the reliability of their output according to certain quality criteria defined by the system's operators. The approach makes use of knowledge graphs that incorporate various ontologies to document, classify and distinguish reliable from non-reliable information processed by an AI system and model an accountability plan that helps an operator to estimate / evaluate the quality of the system's output according to predefined criteria.

While the approach is interesting, the paper comes with a major epistemological deficiency right from the beginning. The concept of accountability in this paper is simply used in a misleading and even erroneous manner: Machines can never be held accountable for something, just natural (humans) or (to a limited extent) judicial persons (organizations) can be held accountable for an action or the consequences arinsing from it.  What is being labeled here as a definition or explanation of accountability, are measures to exercise accountability, but it is not a valid definition in a scientific sense. Hence it is simply not clear what kind of "subject of accountability" the refers refer to. This becomes especially obvious in section 4 and 5. Also, given that the authors do not provide a debatable reference for accountability leads to the impression that the concept is used in a non-scientific, trivial manner.
Of course it is possible to develop something like "accounatbility systems" , which refers to a systemtic / holistic approach to hold actors accountable for something (i.e. this approach exists in the education sector or public health sector with the purpose to govern multi-agent systems where various institutional logics collide). But again, such an approach is not transferable to the research area described in this article. At least the authors do not indicate or refer to such an approach. Hence, the output of a system can never be "accountable" but at best "reliable".

The problem described above becomes even more obvious wehen looking at the related works section. The authors refer in reference 1 to a paper provided by ACM that defines accountaibility in algorithmic systems. If you read what ACM defines as acountability, this definition does not comply with the concept of accounatbility used in ths paper. According to ACM accountability is defined as: "Institutions should be held responsible for decisions made by the algorithms that they use, even if it is not feasible to explain in detail how the algorithms produce their results."

Although the methodological sections of the paper are solid and deserve attention and discussion, the problem stated above simply disqualifies this paper to be accepted. The problem is not just a terminological one, the poblem affects the epistemologiocal foundations of the paper.

A solution to the problem could be to adjust what the authors mean with "AI system". In the paper they define AI systems as: "For the purpose of this paper, the term AI system refers to software comprising `core AI' components (e.g. a machine learning model) and other supporting functions (e.g. API wrappers) [19] allowing it to function either as a standalone solution, or as a part of a larger system." This definition is just to a limited extent compatible or even out of scope of the accountability research. If they define it as a socio-technical system similar to an education system of Health system then their approach becomes debatable. But this would also require a different line of argumentation as given in this paper and probably also a much broader governance approach that transcends the area of knowledge modelling.




**Anonymity:**

No, I would like my review to be deanonymized.

**Reuse And Availability:**

3: Medium

**Strong Points:**

- Relevant topic
- Interessting approach but it remains open whether the formalistic, representational modelling approach is a viable means to finally secure reliability of the system's output and in consequence allow accountable actors to behave responsibly.

**Subreviewer:**

I submitted this review.

**Weak Points:**

The article comes with severe epistemological / conceptual flaws with respect to the concept of "accountability"

---

> ### Author Rebuttal · Authors · 2021-01-29
>
> We thank the reviewer for their comments.
>
> We acknowledge the issue raised about the ACM definition of accountability. Our intent was to use the citation to highlight the wider principles that our framework aims to support. This includes transparency of AI systems that we perceive as a key enabler of processes that may determine accountability.
> In our manuscript we provide several relevant citations covering the concept of accountability, but now recognise that these were not explicitly linked to the discussion of accountability on page 2 of the paper. The sentence beginning “Moreover, …" should now read as follows:
> “Moreover, by accountability, we mean the ability to inspect, review or otherwise interrogate an AI system with the goal of (i) making processes associated with each of its life cycle stages transparent [1, 6, 7, 14, 24, 25]; (ii) demonstrating compliance with hard laws (i.e. laws and regulations), and soft laws (i.e. standards and guidelines) [14, 24]; and (iii) aiding investigations into the cause(s) of failure or erroneous decisions and supporting the identification of responsible parties [1, 6, 7, 14, 24].”
> As a result, our interpretation of accountability (and our framework) is broad in order to encompass the different perspectives presented by these individual references. We also do not claim that our solution alone would address the issue of accountability in AI systems.
>
> Furthermore, in our work we acknowledge and agree with the reviewer that the machine itself is not the one to be held to account and hence focus on recording information about the AI system life cycle to support identification of responsible human actors.
>
> With regard to the issue of scope for our definition of  AI system, we indeed consider such a system to be the "software comprising `core AI' components (e.g. a machine learning model) and other supporting functions (e.g. API wrappers) [19] allowing it to function either as a standalone solution, or as a part of a larger system”; however, as should be evident in our approach - we do not exclude information about events (e.g., decisions) which are part of the wider socio-technical context.
> We also find the suggestion of broadening the scope to include governance approaches interesting, however, we consider this to be out of scope for this work.
>
> Finally, we were surprised by the very low score for reuse and availability given that all evaluation resources, prototype software, and ontologies are available in online repositories linked to from the paper.

---

> > ### Comment · AnonReviewer5 · 2021-02-01
> > **Given the authors' rebuttal I have changed the overall score from weak reject to weak accept.**
> >
> > Given the authors' rebuttal I have changed the overall score from weak reject to weak accept.

---

### Decision · Program_Chairs · 2021-02-23

**Decision:**

Accept

**Comment:**

The reviewers have assessed this paper as addressing a relevant topic with a novel approach. They particularly appreciated that the paper follows a solid methodology for requirements elicitation and ontology design and that it makes many artifacts available. At the same time, the reviewers have identified a few conceptual flows, limitations in  positioning wrt related work. They would have also liked to see a more extensive evaluation that sheds light onto  the approach's  generality, coverage, usability, fit for purpose.

The authors' rebuttal has addressed several of the reviewer concerns, thus swaying the reviewers towards accepting the paper. After discussing the paper extensively in the PC, it was agreed that the merits of the paper outweigh its limitations, and therefore the paper should be accepted with the expectation that authors address the limitations identified by the reviewers as much as feasible.